# Mechanical and Moisture Barrier Properties of Epoxy–Nanoclay and Hybrid Epoxy–Nanoclay Glass Fibre Composites: A Review

**DOI:** 10.3390/polym14081620

**Published:** 2022-04-16

**Authors:** Necar Merah, Farhan Ashraf, Mian M. Shaukat

**Affiliations:** 1Department of Mechanical Engineering, King Fahd University of Petroleum and Minerals, Dhahran 31261, Saudi Arabia; mshaukat@kfupm.edu.sa; 2Interdisciplinary Research Center for Advanced Materials, King Fahd University of Petroleum and Minerals, Dhahran 31261, Saudi Arabia; farhan.ashraf@kfupm.edu.sa

**Keywords:** epoxy, nanoclay, nanocomposite, hybrid glass fibre glass epoxy–nanoclay composite

## Abstract

Epoxy clay nanocomposites have been proven to have improved mechanical, thermal and physical properties over pristine matrix. Thus, the fields of application of epoxy–clay nanocomposites along with their hybrid glass/carbon fibre reinforced composites have grown tremendously during the last few decades. The present review paper covers the research work performed on epoxy clay nanocomposites. It includes the influence of the processing techniques and parameters on the morphology of the nanocomposite, the methods of characterization and the effects of adding nanoclay on the mechanical and physical properties of composite. The improvements in the liquid barrier properties brought about by the addition of nanoclay platelets to epoxy resin are discussed. The variation of physical and mechanical properties with nanoclay type and content are reviewed along with the effects of moisture uptake on these properties. The advances in the development, characterization and applications of hybrid glass fibre reinforced epoxy–clay nanocomposites are discussed. Findings of the research work on the influence of nanoclay addition and exposure to water laden atmospheres on the behaviour of the hybrid glass fibre epoxy–nanoclay composites are presented. Finally, the potential health and environmental issues related to nanomaterials and their hybrid composites are reviewed.

## 1. Introduction

The market of nanocomposites and polymer-based nanocomposites has been growing during the last few decades. The global nanocomposite market size was estimated to be $4.32 billion in 2019 with the expectation to grow $14.37 billion by 2027 [1]. Other sources [2] estimate that the global nanocomposites market size is expected to be $7.48 billion by 2022, expanding at a compound annual growth rate (CAGR) of 6.4%. This is driven by increasing demand for high volumes of nanocomposites in various industries such as automotive, construction and semiconductors. The market for polymer nanocomposites was about $7 billion in 2018 with a growth projection of 21% by 2025 [3]. Of these nanocomposites, polymer/nanoclay is projected to represent 17% of CAGR and reach $4.2 billion by 2027 [4]. This growing demand for polymer nanocomposite materials and more specifically for polymer nanoclay has attracted a large number of researchers to study the effects of adding nanoclay to various polymer resins and mainly epoxy. Anadão [5] reported that the first mention of polymer-nanoclay (PCN) composites was in 1949 by Bower [6]. Pavlidou and Papaspyrides [7] described the important milestones of polymer layered silicates since it was first introduced in 1949 to 2008. However, PCN became technically viable during the period following the demonstration of improving properties of polymers by adding nanoclays by Toyota researchers [8]. In 2007, Okada and Usuki [9] mentioned that so far, only nylon–clay nanocomposites are used for practical applications, but other PCN are becoming increasingly useful.

A large number of review papers on nanocomposites have been published in the last three decades [7,10,11,12,13,14,15,16,17,18,19,20,21,22,23]. Hussain et al. [10] published a comprehensive review of research related to polymer nanocomposites. Asif et al. [16] summarized research underway related to epoxy clay nanocomposites. Kamal et al. [13] reviewed some of the common fabrication techniques of polymer-based nanocomposites. They described the principles of each technique detailing the advantages and disadvantages.

The present review focuses on epoxy–nanoclay (E-nc) composites: their structures, mechanical, thermal and physical properties. The resistance of the nanocomposite to water uptake and the variation of mechanical and physical properties with moisture diffusion is discussed. The research work on glass fibre epoxy–nanoclay (GFRE-nc) hybrid composites is reviewed in terms of their fabrication methods, structures and mechanical properties. The last section discusses the possible health and environmental impacts of PNC and GFRE-nc composites. 

## 2. Epoxy Nanoclay Composites

### 2.1. Epoxy

Epoxy resin is a class of reactive prepolymers that comprise a group of cross linkable materials. The characteristic group form a three-member ring structure know as epoxy, epoxide, oxirane, glycidyl or ethoxyline group as shown in Figure 1.

Due to higher reactivity of epoxy group, it can react with a wide range of curing agents. These curing agents are generally of two types: (i) catalysts or (ii) hardeners. Catalysts are usually from tertiary amines, and they are used in low concentration below 1%. In most cases, catalysts are used with hardeners to form resin. Aromatic and aliphatic amines and carboxylic anhydride are common examples of hardeners. Aliphatic amines are normally cure at room temperature whereas aromatic amines and carboxylic anhydride are used for curing at higher temperature. The properties of cured epoxy resin vary depending on the chemical structure of curing agents and curing condition. Generally, resins cured at higher temperature develop improved properties such as strength, stiffness and high glass transition temperature compared to room temperature cured resin [24]. In terms of general properties, epoxy resins have excellent heat and chemical resistance, high tensile and adhesive strength, good hardness and impact resistance and electrical insulation [16,25]. 

### 2.2. Clays

Clays are aluminium or hydrous silicates containing several cations such as oxygen, hydroxyl and silicon. These cations are organized in two-dimensional sheet-like structures. Clay minerals are also referred as layered silicates because they comprised of 1 nm thick silicate layers containing alumina and silica sheets joined together in various proportions and stacked on top of each other with interlayer distance. Clay minerals are classified as dimorphic (two-sheet minerals), trimorphic (three-sheet minerals) and tetramorphic (four-sheet minerals) based on condensation ration of silica to alumina sheet [26]. 

### 2.3. Nanoclays

Nanoclays are sheets of nanoparticles made of layered mineral silicates. They are classified in different categories depending on their nanoparticle morphology, and chemical composition. Commonly used nanoclays for epoxy resins are organically modified montmorillonite clays with filling amounts ranging from 1% to 10%. Organically modified montmorillonite clays, which are naturally occurring inorganic materials, belong to the smectite clays. These layered silicate clays maintain their two-dimensional (2D) crystallographic plane even after mixing with polymers. Figure 2 depicts the 2D structure where a central octahedral sheet of alumina is joined to two external silica tetrahedral resulting in oxygen ions of the octahedral sheet becoming part of tetrahedral sheets [7,27]. The clay layer is approximately 1 nm in thickness and its other dimensions are from 100 to 1000 nm resulting in high aspect ratios high aspect ratios which favour its reinforcing and barrier capabilities. The chemical composition of ½ crystalline unit cell is $Na_{0.33}$($Al_{1.67}-{\mathrm{Mg}}_{0.33})Si_{4}O_{10}{\left(\mathrm{OH}\right)}_{2}$ [7]. The composition of the clay particles (ionic and polar in nature), in general, cause them to be hydrophilic i.e., incompatible with organic polymers. Therefore, it is necessary to modify their surface to organophilic prior to their use. Organic ions such as ammonium or phosphonium are commonly used to modify the clay minerals [7,16,17,28,29,30,31,32,33]. This organic modification increases the interlayer spacing (d-spacing) between the nanoclay platelets to about 2 nm. This d-space expansion is expected to allow the diffusion of polymers into interlayer galleries.

When organically modified nanoclays are mixed with polymers, they generally result into one or a combination of two of the three main composites categories [35]: (1) conventional composites, where the basal spacing between clay layers does not increase during mixing and the polymer matrix does not penetrate into the clay galleries (Figure 3A); (2) intercalated nanocomposites, where polymers penetrate between the ordered layers of clay resulting in an increase in interlayer spacing (Figure 3B); and (3) exfoliated nanocomposites, where individual nanometre-thick layers of clay are separated and distributed within polymer matrix (Figure 3C).

If d-spacing for clay layers is larger than 5 nm and transmission electron microscopy (TEM) can detect the disordered arrangement, then the polymer-clay nanostructure is called exfoliated nanocomposite [28,29,30]. A fully exfoliated structure would have an interlayer spacing greater than 8 nm. The exfoliated morphology has been shown to maximize the interactions between polymers and nanoclays, resulting in an appreciable increase in thermal, mechanical and barrier properties of the resin. Several researchers have described how these structures form and the type of 2-D silicate clays and their different modifications [7,18,27,36]. They explained the role of different modifiers in altering interfacial interactions and thus affecting the nanocomposite properties. The clay and polymer type determines the nanostructure of the composite. Pavlidou and Papaspyrides [7] explained that the longer surfactant chains will force clay layers further apart. This explanation was mainly based on the work of Wang et al. [37] who prepared organoclays with different alkylammonium chain lengths and also used an organophilic clay modified with methyl tallow bis-2-hydroxyl quaternary ammonium, (Cloisite 20A) which has two long alkyl chains to find that as the alkylamine chain length increases, the interlayer spacing increases. Xidas and Triantafyllidis [38] synthesized nanocomposites using EPON 828RS epoxy and five different types of montmorillonite clays; two modified with octadecyl ammonium ion (Nanomers I.28E, I.30E) and three Closites (C10A, C15A and C20A). From the five organoclays used, they found that only Nanomer I.30E provided exfoliated nanostructures. The authors attributed this to the catalytic function of the acidic primary onium ions that enhanced intergallery polymerization rate of I.30 E. Al-Qadhi and Merah [39] compared the effects of Nanomers I.28E and I.30E and Cloisites C10A and C15A on the structure and properties of the epoxy–nanoclay composites finding that samples made with I.30E clay showed improved dispersion of nanoclay in DGEBA epoxy matrix. TEM analyses revealed that I.30E nanocomposites had mainly disordered intercalated with some exfoliated morphology while the structures of nanocomposites prepared with C10A and C15A were dominated by ordered intercalated morphologies with lower basal spacing. 

In a study by Al-Qadhi et al. [29], it was reported that the addition of 10 wt% of I.30E nanoclay reduced the glass transition temperature (T_g_) of DGEBA epoxy from 152 °C to 135 °C. This was because of less crosslinking density between clay layers within clay tactoids. The same phenomenon is reported by other researchers [38,40]. One reason for this reduction in T_g_ was given as increasing concentration of long-chain modifier in the matrix [38]. Another reason behind this phenomenon is the changed stoichiometry between epoxy and curing agents [40]. Some studies have shown improvement in T_g_ [41,42], on the other hand, some others reported no change on T_g_ [43]. The types of nanoclays are dictating the increase or decrease in T_g_ [38]. 

### 2.4. Processing Methods and Techniques

Without proper dispersion and distribution of the nanoclay in a polymer matrix, the advantage from the high aspect ratio of the clay platelets is compromised. In fact, clustering of the clay particles can act as defects resulting in adverse effects [44]. That is why it is essential to use proper processing and curing methods with optimized parameters.

In general, there are three methods to mix and disperse nanoclays in polymers: (1) direct mixing of polymer and nanoclay, (2) in situ polymerization in presence of nanoclays and (3) in situ formation of the nanoparticles and in situ polymerization. The first technique is used for thermoplastics. In a second technique, nanoclays are dispersed in the monomer or monomer solution, then polymerized by standard polymerization methods. This is the preferred method for processing epoxy–clay nanocomposites as it has the potential to graft the polymer onto the particle surface. The third method is employed to fabricate hybrid nanocomposites because it allows the mixing of the two phases. Hussain et al. [10] and Pavlidou and Papaspyrides [7] have provided excellent reviews of the processing methods and techniques used in polymer-clay nanocomposites. Others [11,45,46,47,48,49] described the dispersion techniques employed to synthesize epoxy–nanoclay composites. Among the general methods for clay dispersion in the epoxy resin reported by Zabihi et al. [11] are mechanical stirring [45], ultrasonication [46], high shear mixing [47,48,49], high pressure mixing [45] and slurry processes [50]. Furthermore, and in order to improve the dispersion of nanoclays into polymer matrices, Al-Qadhi et al. [29] combined high shear mixing and ultra-sonication, finding that ultrasonication did not bring about any improvement in the dispersion of nanoclays in the epoxy matrix. Other researchers [51,52] employed solvents in order to increase the efficiency of these methods and to make more homogenous dispersions of nanoclays into polymer matrices. Brown et al. [51] and Liu et al. [52] used acetone as a solvent and found that the solvent facilitated the mixing process by reducing the viscosity of the system. Each of these techniques has several parameters that need to be controlled to arrive at an efficient mixing process. Al-Qadhi [30] and Xidas and Triantafyllidis [38] showed that nanomers disperse easier than some closites and that the lower the clay loading, the better the dispersion. What works for low nanoclay content may not work for high clay content. Optimum clay loading depends upon clay and epoxy type and process parameters [43,53,54,55]. An optimization process has to be followed for each type of polymer, clay and clay loading. For example, high friction processing requires the control of mixing speed, mixing time and slurry temperature. Using a three roller high shear mixer, Al-Qadhi et al. [47] and Al-Qadhi and Merah [56] showed that I.30E nanoclays are well dispersed in Diglycidyl ether of bisphenol A (DGEBA) epoxy matrix at 6000 rpm for 60 min. 

Once the mixing is completed curing agents have to be added and here the process has to be optimized. The selection of the right curing agent and the right quantity is a critical task as it determines the structure and properties of epoxy–clay nanocomposite [16]. Along with the curing agent, several other factors are to be considered during curing such as degassing time and temperature, as well as curing time and temperature [47,57,58]. Pre-curing time and temperature have a significant effect on crosslinking of samples [57]. The degree of crosslinking increases with an increase in curing time and temperature. However, the effect of pre-curing temperature is found to vanish when post-curing at high temperature is used [57]. With regards to degassing, some efforts [58] have shown that increasing degassing temperature and time can enhance the diffusion of epoxy into intergallery of nanoclay platelets. 

As a general practice, amine and anhydride-based curing agents are used in epoxy clay systems [16]. Anhydride as a curing agent develops exfoliated morphology because it is a liquid and can easily diffuse into clay gallery [59,60]. Contrarily, amine-based curing agents such as diamino diphenyl methane (DDM) impart intercalated morphology as it exists in a solid state [59,60]. Despite type of curing agent, the diffusion rate and reactivity of the curing agent also play a role in the exfoliation of clay. For instance, aliphatic amine-cured epoxy produced exfoliated morphology compared to cycloaliphatic amine-cured epoxy, due to higher reactivity [61]. However, the concentration of curing agents in epoxy along with their reactivity also determines the morphology. For example, when DGEBA epoxy with octadecyl ammonium modified MMT is cured with less than a stoichiometric amount of meta-phenylene diamine (MPDA), an exfoliated nanocomposite is formed. However, if a higher concentration MPDA is used then it results in intercalated nanostructure [62]. A higher concentration of curing agent promotes extragallary cross linking and intercalated nanocomposite. 

Some efforts [63] also studied the exfoliation behaviour based on electronegativities differences of aromatic diamine and curing temperature. Epoxy clay nanocomposite containing DGEBA/PDA and DGEBA/MDA systems promote intercalated morphology due to higher reactivity of curing agent. Whereas DGEBA/DDS system produces exfoliated structure due to lower reactivity of curing agent. The morphology of epoxy–clay nanocomposite has a prominent effect on their mechanical properties. For instance, intercalated structure has higher toughness whereas exfoliated structure based on anhydride cured DGEBA epoxy possess higher stiffness [64].

The right combination of all of the mixing and curing factors is hard to come about. The degree of exfoliation depends on all of these and more. Therefore, optimum values for these parameters should be determined in order to produce the desired structure of epoxy–clay nanocomposite. 

### 2.5. Characterization of Epoxy–Clay Nanocomposites

As discussed earlier, the degree of dispersion of nanoclays in polymer resin determines enhancement in physical, thermal and mechanical properties of the PNC. Though huge advances are made in the fields of nanotechnology and nanoscience, researchers in the area of nanomaterials are still facing challenges in terms of the availability of proper instruments for observing, measuring and manipulating these materials at the nanometre level. Therefore, proper characterization techniques are crucial for understanding how the resulting polymer-clay morphologies affect the nanocomposites physical, thermal and mechanical properties.

Several characterization techniques [7,10,28,65,66] have been employed for epoxy nanocomposite characterization. Among those, X-ray diffraction (XRD), scanning electron microscopy (SEM), transmission electron microscopy (TEM), thermal gravimetric analysis (TGA) and dynamic mechanical analysis (DMA) are the most common techniques. These techniques are summarized in terms of how they have been used to characterize epoxy nanocomposite.

X-ray diffraction (XRD) is one of the commonly used techniques to find the type of morphology structure and measure the interplanar spacing (d-spacing) between clays using Braggs law. Several researchers [30,46,47,48,58,67,68,69,70,71,72,73,74,75] have used XRD to characterize clay dispersion in epoxy nanocomposite. In parallel to its easy access and availability, this technique has several limitations. XRD can only detect the periodically stacked montmorillonite layers; disordered or exfoliated layers cannot be observed and no peak appears for this structure [47]. Hence, only the intercalated structures with individual silicate layers 7 nm separated appear as a peak in XRD. Generally, two types of scattering techniques have been employed to characterize epoxy nano composites. Small-angle X-ray scattering (SAXS) is the first one that measures scattering intensity at scattering angle of 2θ, where θ is the diffraction angle. The value of θ for SAXS ranges between 0° and 5°. Wide angle X-ray diffraction (WAXD) is a second scattering technique that measures the range of large scattering angles. During XRD, most clays show peaks between scattering angles of 3° to 5° [67]. Ngo et al. [45] measured the d-spacing of C30B clay powder and nanocomposites with larger diffraction angles in the range of 2° to 10°. His study [45] showed that the C30B clay powder has one peak at 4.8° corresponding to d-spacing of 1.85 nm. On the other hand, nanocomposites show peaks at about 2.3° corresponding to d-spacing of 3.8 nm. The later finding revealed that the resultant nanocomposites have intercalated morphology.

SEM has been used to study the microscopic view of clay dispersion. Several researchers [58,68] used scattered nodular features to characterize nano clay filled epoxy. These nodular features represent cluster of agglomerated clay. They showed that the size of agglomerated clusters decreases with increasing mixing speed. Al-Qadhi et al. [58] also studied the effect of mixing time on agglomerated clay clusters. They showed that longer mixing time results in small clusters. Wang et al. [68] studied the crack initiation in the E-nc using SEM images. They explained that clay layers act as a stress concentrator and promote several microcracks. Macrocracks were also found at the interfacial due to differences in the inside and outside physical properties of clusters [68]. In addition to the above characterization, high resolution SEM has been used to study the adhesion between fibre and matrix [76]. Study [76] showed that higher percentage of reinforcement in epoxy creates lesser space in matrix and provides better adhesion. 

Transmission electron microscope is an imaging technique used to obtain a direct view of the dispersion state in materials. This technique has been widely employed to study the morphology of epoxy–clay nanocomposite at the sub-microscale [63]. Figure 4 shows the exfoliated and intercalated structures of epoxy–clay nanocomposite. The d-spacing for disordered intercalated morphologies can also be estimated with TEM [58]. The higher magnification of TEM imaging provides information about phase separation in epoxy cured nanocomposite, whereas low magnification shows the separation of silicate monolayers [65,66]. 

### 2.6. Water Uptake

The water uptake strongly affects the properties of epoxy resins; in fact, water laden atmospheres at high temperatures are detrimental to polymers. Due to the plasticizing effect of water on cured epoxy polymer, the glass transition temperature, of the systems is, in general, reduced by 20 °C for each 1% moisture uptake [25,77]. The existence of molecule-sized hole in epoxy matrix structure and the polarity between water and epoxy group drives the water uptake [29]. Many different methods have been used to study the effect of moisture on epoxy composites and found it to negatively affect the performance of the resin. Compared to unaltered polymer, epoxy reinforced with nanoclay offers enhanced properties, especially exfoliated structures. Among these properties is the nanoclay ability to reduce moisture uptake. 

As mentioned in the introduction, the epoxy-based nanoclay composites with high moisture and gas barrier properties make them attractive for applications such as vehicle parts and matrices for fibre-reinforced plastic structures used in wet environments. Moisture uptake in epoxy–clay nanocomposites has attracted a lot of attention. A number of researchers [55,78,79,80,81] have shown that by adding nanoclay in epoxy matrix, its hydrophobicity is improved by reducing both the diffusivity rate and the maximum water uptake. Kim et al. [55] and Liu et al. [79] reported that water uptake of epoxy clay nanocomposites decreased with the increase in clay content. Al-Qadhi et al. [29] have observed similar behaviour of moisture ingress in epoxy clay nanocomposite. They showed that the addition of 1.0 wt% of I.30E nanoclay to epoxy resulted in a decrease of more than 50% in its diffusivity; from 2.7 × 10^−7^ to 1.37 × 10^−7^ mm^2^/s. As shown in Figure 5, the maximum water uptake, in terms of percent weight gain, decreased almost linearly with clay content; from 2.2% for neat epoxy to 1.7% for nanocomposite containing 5.0 wt% of I.30E [29]. Bagherzadeh et al. [71] showed that a nanoclay-epoxy composite coating with 1.0 wt% clay loading resulted in 70% reduction in water uptake. Other researchers [47,82] reported reductions by up to 40% of moisture absorption as compared to pristine epoxy resin. In another study, water permeability was observed to decrease by 80% with as little as 5.0 vol% of nanoclay [57]. The barrier properties depend also on the type of organoclays and mainly on the basal distance between the silicate layers used as nanofillers. Kim et al. [55] studied the resistance to water permeability of three epoxy nanocomposites prepared with three organoclays (KH-MT-TJ2, C20A, C20A and I.30P) and found that I.30P offered the highest resistance to moisture uptake. This was because of the observed I.30P larger d-spacing and better dispersion. Al-Qadhi and Merah [39] compared the diffusivity rate and maximum water uptake of samples containing 1.0wt% of I.28E, I.30E, C10A, C15A and C20A and found that the highest reduction in maximum water uptake was that of I.30E clay (6%) and the lowest was for the samples containing I.28E clay (2.5%). These findings agree with what was reported by several other researchers [7,29,37,38]. The decrease in the diffusivity and water uptake is related the tortuosity effect created by the presence of dispersed clay platelets with high surface area. The water molecules have to move around the exfoliated/intercalated clay layers during diffusion in the nanocomposites.

It is, however, observed (Figure 4) that nanocomposites containing larger amounts of clay (10.0 wt%) showed less change in diffusivity rate and maximum water uptake [29]. The above-mentioned increase in clay clusters and microvoids at high clay loadings are responsible for this reduction in the resistance to water permeability. The reason is that in place of moving around each nanolayer, water molecules move around the clay aggregate. The difficulty in mixing and degassing of nanoclay and epoxy due to increase in viscosity of the mixture leads also to increases in microvoids. This lack of proper mixing and dispersion will be shown to also have negative effects on the mechanical properties. 

### 2.7. Mechanical Properties

Nanocomposites consisting of epoxy and organically modified nanoclays often show better mechanical properties than those of pristine epoxies. A number of studies reported improvements in mechanical, physical and thermal properties of epoxy because of adding clay [38,41,83,84], on the other hand some researchers reported no such improvements [48,85].

Zabihi et al. [11], in their technical review on epoxy–clay nanocomposites developed a table in which they summarized the different effects of the type of clay modifier and the method of dispersion on the structure of the nanocomposite and its mechanical properties. The results taken from more than 20 publications show that, in general, while the tensile modulus and fracture toughness increase with increasing clay loading, the tensile strength improvement is dependent on the structure of the nanocomposite. If the structure is exfoliated or highly exfoliated, the tensile strength is higher than that of the neat epoxy [84,86,87,88,89,90] but if it is intercalated or exfoliated/intercalated, the improvement depends on the clay loading. The largest increase in the tensile strength was reported by Jlassi et al. [88] to be more than 68% for 0.1 wt% of octadecylammonium (I.30E) polyaniline clay loading where the highly exfoliated nanocomposite was prepared by sonication. The variation of the tensile strength with clay loading; rising at low contents and decreasing with higher clay loadings have also been reported by Al-Qadhi et al. [29], Xidas and Triantafillidis [38], Al-Qadhi and Merah [35], Daniel et al. [49] and Kusmono et al. [28]. Akbari et al. [15], however, found an improvement in tensile strength with increasing clay loading. The fracture strain is found to decrease almost linearly with increasing clay loading up to 5.0 wt% [3]. Kusmono et al. [91] found that addition 3 wt% of nanoclay in epoxy resulted in the highest tensile, flexural and impact strengths as well as fracture toughness. Merzah et al. [92] studied the effect of treated nanoclay loading on the mechanical properties of epoxy and found that the tensile properties were improved by the addition of small amounts of nanoclay. They reported that the optimum clay loadings were 2.0 wt% for tensile strength and maximum load while the percentage elongation and Young’s modulus increased gradually with clay loading. 

Al-Qadhi and Merah [39] found that while 2.0 wt% of I.30E and C15A clay did not have any effect on the tensile strength, the same amount of I.28E and C10A reduced the strength by more than 30%. They explained this difference by showing that the nanocomposites prepared with I.30E and C15A had more exfoliated structure with larger d-spacing than those prepared with I.28E and C10A. Furthermore, SEM images of fractured surfaces revealed the presence of more clay clusters and microvoids in I.28E and C10A samples. Clustering of clay platelets and microvoids are known to become sites of stress concentrations, leading to lower mechanical strength. Table 1 shows a comparison of the tensile properties of nanocomposites prepared with 1.0 to 3.0 wt% of the above-mentioned organically modified clays (I.28E, I.30E, C10A and C15A). It can be seen that, while the average value of the tensile strength of I.30E nanocomposite increased or remained constant, that of I.28E and the Closites (C10A and C15A) decreased. On the other hand, there is a clear enhancement in the modulus of elasticity all types of nanocomposites. This increase in tensile modulus can be attributed to the high stiffness of dispersed nanolayers of clay in epoxy resin, which restrict the mobility of polymer chains during tensile loading. The improvement is stiffness of nanocomposites is also due to the good interfacial adhesion between the clay particles and the epoxy matrix, especially in exfoliated structures. As the stiffness increases, the nanocomposite becomes more brittle resulting in lower fracture strain as can be seen in Table 1.

In addition to the size of the organic modifier and its compatibility with the monomer, the clay content and the mixing techniques are known to affect the tensile properties of the nanocomposites. Figure 6 illustrates the variations of the tensile strength, modulus of elasticity and fracture strain of DGEBA epoxy nanocomposites prepared with I.30E clay contents, ranging from 1.0 to 10.0 wt% [29]. It can be seen that for clay loadings above 1.0 wt%, the tensile strength decreased almost linearly with clay content until 5.0 wt%. The improvement in the tensile strength at low clay loading is the direct result of the observed exfoliated morphology of the nanoclay. Similar to what was reported earlier, the tensile modulus increased and the ductility decreased with clay content. Table 2 illustrates the effects of clay loading (1.0 to 10 wt%) and mixing techniques on the tensile properties of epoxy nancomposites prepared with the same type of organically modified nanoclay I.30E. It can be seen that, in general, high shear mixing and low clay contents resulted in better tensile properties. The work of Qi et al. [75] included also the influence on I.30E nanoclay content on the fracture toughness of epoxy. They found that the fracture toughness was improved by the addition of clay. The highest improvement, about 35%, was obtained for 5 wt% I.30E clay content. As explained by a number of researchers [11,29,39,84,86,87,88,89,90], the improvement in tensile strength and other mechanical properties depends mainly on the morphology of the nanocomposite. The more exfoliated the nanocomposite structure, the better the improvements in the mechanical, thermal and physical properties.

As reported above, exposure of epoxy to moisture is seen to degrade its glass transition temperature (T_g_) by about 20 °C for each 1% moisture uptake [25,77]. The plasticizing effect of water on cured epoxy polymer which results in the reduction of glass transition temperature has similar effects on the mechanical properties of the polymer. Al-Qadhi and Merah [35] and Al Amri and Low [93], found that water absorption in epoxy reduced the flexural strength and modulus of the nanoclay-epoxy composites. They, however, observed that the addition of nanofillers enhanced these flexural properties as well as both fracture toughness and impact strength when compared to wet pristine epoxy. Al-Qadhi et al. [94] found that the tensile strength and the elastic modulus of neat epoxy and its nanoclay composites were adversely affected by moisture. Similar to glass transition, the effect on the tensile properties was found to be proportional to the amount of liquid sorption. The reductions in strength and modulus due to water uptake for the nanocomposites containing 1.0 wt% of I.30E nanoclay were about 1/2 of that of neat epoxy. They, however, observed an increase of about 26% in ductility and 42% in fracture strain due to water uptake [94]. They concluded that there may be a competitive effect due to stiffening introduced by clay layers and plasticizing of the matrix because of water molecules diffusion.

## 3. Hybrid Glass Fibre Reinforced Epoxy Nanoclay Composites

The use of glass fibre reinforced epoxy (GFRE) is increasing to manufacture pipes for potable water, seawater, sewage and other corrosive fluids. One of such application is the use of GFRE pipes in water production and transportation networks. Although GFRE offers excellent mechanical strength under both static and cyclic loading conditions, one major obstacle to a wider use of GFRE in piping systems is the above shown affinity of epoxy resin to moisture absorption [25,29,77]. The absorbed moisture is known to swell, crack and plasticize epoxy resin and degrade the fibre/matrix bonding and thus significantly lower the mechanical strength of the composite. The reinforcing of neat epoxy resin with glass fibre has been shown to improve its strength. The long term durability of GFRE is, however, affected negatively when exposed to water laden environments and especially at high temperatures [95,96,97,98,99,100]. Water laden atmospheres combined with high temperatures and other environmental conditions have long been regarded as the most damaging environments for polymer based composites [95,96,97,98,99,101]. 

Thus, any material modification strategy, which can lead to lower moisture absorption in GFRE composites can be extremely helpful in expanding the applications of GFRE composites to moisture laden service conditions. One such strategy is to generate tortuosities in the path of absorbed liquid molecules to limit their diffusion into the matrix material and along the fibre/matrix interfacial pathways, which can be achieved by complete exfoliation of a filler such as nanoclay into the matrix. As demonstrated above, the complete exfoliation of fillers in polymer matrices remains quite a challenge for researchers. Another dimension is added to this challenge when a nanocomposite is used as matrix for GFRE. 

As explained above, adding nanoclay to the neat epoxy has, in general, helped reduce its moisture uptake as well as enhance some of its physical, thermal and mechanical properties. The combination of the nanoclay together with glass fibres as reinforcements in a hybrid nanocomposite was the next step that researchers took during the past decade [41,43,53,102,103,104,105,106,107,108,109]. Avila et al. [103] investigated S2-glass/nanoclay/epoxy laminated composite where 1.0 wt% to 10.0 wt% nanoclay were mixed with epoxy by stepwise direct mixing and the hybrid composite is fabricated using the conventional stacking sequence and vacuum assisted wet lay-up lamination. The resulting S2-glass/epoxy–nanoclay composite is a laminate with 16 layers and 65% fibre volume fraction. The authors found that the addition of nano-sized clays increased the composite impact strength by up to 50% at 10 wt%. Lin et al. [53] studied layered silicate/glass fibre/epoxy hybrid nanocomposite prepared by vacuum-assisted resin transfer moulding (VARTM). They used three methods to mix 1.0 to 5.0 wt% nanoclay with epoxy: direct mixing, solution mixing and ultrasonication. The mixture was injected in steel mould containing five layers of unidirectional glass fibres stacked in the longitudinal or transverse directions. The characterization of the hybrid nanocomposite showed that: an intercalated structure of the clays resulted from sonication, good impregnation was obtained when unidirectional glass fibres were aligned in the longitudinal direction and mechanical and thermal properties were enhanced by the introduction of a small amount of organoclay to the glass fibre/epoxy composites. Dharmavaram et al. [104] used resin transfer moulding (RTM) to manufacture nanoclay/glass/epoxy composites with 2, 5 and 10 wt% of nanoclay and 14% glass fibres. They observed higher filtration with increasing nanoclay content. The problem of nanoparticle filtration was also observed by Lekakou et al. [105] in RTM through 10 layers of woven fabrics at a fibre volume fraction of 50%. They found that Laminates of lower fibre fraction were produced successfully by Resin Infusion under Flexible Tool (RIFT). It is revealed through three-point bend tests and Dynamic Mechanical Thermal Analysis (DMTA) that GFR with nanoclay reinforcement showed higher Young’s modulus and interlaminar shear strength. Aktas et al. [106] prepared prepregs of chopped strand E-glass fibre mats using waterborne epoxy resin mixed with nanoclay. They noted that adding water to epoxy reduced clay clustering. Their results showed that inergallery spacing of the layered clay increased leading to enhancement of flexural stiffness by more than 26%, despite a 13% decrease in interlaminar shear strength. Similar results were reported by Bozkurt et al. [43] for non-crimp glass fibre reinforced layered clay/epoxy nanocomposites prepared by hand layup and Norkhairunnisa et al. [107] for chopped strand mat fibre glass clay/epoxy nanocomposite prepared by the same method. Rafiq et al. [108] investigated the effect of adding organoclay I.30E to GFRE composites on its flexural and fracture toughness and concluded that up to 1.5 wt% nanoclay addition improved both properties. Rafiq et al. [109] found that the addition of 1.5 wt% I.30E nanoclay improved the impact peak load by 23% and stiffness by 11% of GFRE. They observed that clay agglomeration in samples with 3.0 wt% loading contributed towards limiting the improvement in impact resistance. Abd El-baky et al. [110] investigated the crashworthiness of hybrid nanocomposites tubes manufactured with plain weave E-glass fabric, epoxy halloysite nanoclay, montmorillonite clay and revealed that GFRE hybrid composite tubes showed high improvements in energy absorption characteristics. Sharma, et al. [111] studied the influence of processing variables, such as the temperature of the mixture, ultrasonication parameters and curing, on the performance of glass fibre reinforced epoxy–clay nanocomposites manufactured by vacuum-assisted wet layup technique. They reported that matrix optimum curing conditions are essential for the improvement of the hybrid GFRE-nc composite. 

Besides the improvement in static properties of the GFRP material, the addition of nanoclay is found to considerably enhance quasi-static and dynamic properties [112] as well as fatigue strength [102] of GFRP. The results of the quasi-static and dynamic experiments performed by Gurusideswar et al. [112] show that the tensile behaviour of E-nc and GFRE-nc composites is dependent on the strain rate. The tensile moduli and tensile strengths tremendously with increasing strain rate. 

Zainuddin et al. [41] observed that the addition of 1–2 wt% of nanoclay to E-glass fibre reinforced epoxy decreased its absorption of water and the addition of 2%wt of nanoclay enhanced its flexural stiffness under all exposure conditions due to enhancement in interfacial bonding. Kornmann et al. [81] reported similar enhancements in flexural properties in glass fibre reinforced epoxy-layered silicate nanocomposites. They explained that improvement in flexural strength and modulus of GFRE-nc samples, prepared by hand layup and vacuum bagging, was due to a stronger epoxy–fibre interface created by the adhesion of silicate layers with glass fibres. The authors [81] reported that higher water uptake was observed in the nanocomposite and the corresponding laminates at 50 °C, compared to water uptake at 23 °C. Similar conclusions concerning the degradation of the flexural properties of GFRE-nc specimens were arrived at by Manfredi et al., [80]. Rafiq and Merah [113] investigated the effects of nanoclay addition and water uptake, at 23 °C and 80 °C, on the flexural properties of GFRE-nc manufactured by hand layup with hot pressing. Their results, as shown in Figure 7, revealed that addition of 1.5 wt% of I.30E nanoclay has resulted in optimum improvement of flexural strength of GFRE; increasing from 209 MPa for GFRE to 232.6 MPa, for GFRE with 1.5 wt% clay loading. Because of the observed clay clustering and presence of microvoids, GFRE-nc prepared with 5.0 wt% resulted in about 5% reduction in flexural strength. The flexural modulus of GFRE-nc showed a similar trend; increasing from 8.30 GPa for GFRE to 9.36 GPa for the hybrid composite with 1.5 wt% of clay, then gradually decreasing for higher clay loadings, reaching 8.47 GPa at 5.0 wt%. The authors [113] found that moisture diffusion rate and maximum water uptake of GFRE decreased with increasing clay loading. Higher water temperature resulted in higher water uptake and moisture diffusion at 80 °C was about 80% higher than that at room temperature. The post exposure flexural tests revealed that the water uptake has in general resulted in lower flexural strengths and stiffness. The addition of nanoclay has reduced the effect of moisture uptake on GFRE flexural properties. Figure 7 shows that even with maximum water uptake at room temperature GFRE-nc with a clay loading of 1.5 wt% has a flexural strength which is higher than that of dry GFRE, but exposure to moisture resulted in the degradation of GFRE-nc flexural strength by about 36% at 80 °C.

## 4. Environmental Impacts of Epoxy–Nanoclay Composites

Understanding potential health and environmental issues related to nanomaterials is an important field of scientific inquiry and it is essential to develop complete picture of nanomaterials and their interaction with environment. Manufacturing of nanomaterials usually requires processes such as chemical and physical vapor deposition, which use large amounts of energy and material inputs and these processes release emissions to air, water and soil [114]. Several researchers have investigated environmental impacts of nanocomposites and findings of these studies can help to identify potential environmental impacts and health hazards resulting from production and use of nanoclay composites. Summary of key studies related to environmental impacts of nanocomposites is presented next. 

Despite the growing use of nanomaterials in various applications, concerns about their negative impact on human body remain [10,114,115,116]. In literature three methods are mentioned by which nanoparticles can enter human body: inhalation, skin contact and ingestion [10,117,118]. Thus far, the major concern is inhalation of nanoparticles during processing, use and disposal of products containing nanoparticles. It has been shown that inhalation of micron-size particles of quartz or asbestos were harmful to human health and same concern remains about nanoparticles [10]. Hence, there is a need for developing protocols and methodologies to investigate toxicology of nanomaterials in various applications [115]. 

In addition to reporting about toxic effect of nanomaterials, several researchers have used Life Cycle Assessment (LCA) to measure environmental impacts of nanomaterials. LCA is a standard methodology that is used to evaluate environmental impacts of a product or a process throughout its entire life cycle, i.e., from extraction of materials to the end-of-life treatment [119,120,121,122]. It identifies the materials, energy and emissions throughout the entire lifecycle of a process and then offer results to point out environmental hotspots and suggestions to improve environmental profile. LCA can also help to prevent shifting of environmental burdens from on stage of lifecycle to the other.

Several studies have used LCA to quantify greenhouse gas emissions for various materials used in nano composites. The values for global warming potential for epoxy and glass fibres are widely reported in the literature. For example, one study [123] has reported that 6.6 kg of CO_2_ is released for production of one kg of epoxy. Similarly, another study reported that 6.6 kg of CO_2_ is release while producing one kg of epoxy [124]. These CO_2_ emissions are primarily generated by energy used during production process for epoxy. Data about CO_2_ emissions while producing glass fibres are also easily available. It is reported that about 1.5 kg of CO_2_ is release while producing 1 kg of glass fibres [125]. Another study reports 2 kg of CO_2_ for one kg of glass fibre production [124]. Majority of these emission are attributed to the production of glass that is the then used to make glass fibres. Due to novelty of process and variation in techniques used to produce nanoclays, CO_2_ numbers related to nanoclay production are not as widely reported. One study estimates that 1522 kg of CO_2_ is released while producing 1 kg of montmorillonite clay [123]. Using these numbers one can estimate CO_2_ emission per kg production of epoxy nanoclay composites. However, in order to fully understand the environmental profile of these materials environmental impacts related to use, transportation and recycling of these materials in various applications should be analysed. 

Lloyd et al. [124] have reported a LCA study on use of nanocomposites in automotive body panels and they concluded that substituting steel body panels with nanocomposite body panels can reduce energy use and environmental discharges over the entire life cycle. In a similar study [125] authors reported that in general nanocomposites do not have higher environmental impacts as compared to conventional materials. Furthermore, if the use of nanocomposites leads to a lightweight component, then life cycle environmental impacts for nanocomposites are less as compared to traditional materials. Nevertheless, they pointed out that more research is needed to investigate the toxic effect of free nanoparticles in the atmosphere. In addition there is lack of understanding about environmental impacts of recycling of nanocomposites [114]. Therefore, future research on LCA of nanomaterials should consider various end-of-life scenarios such as landfilling and recycling. 

In addition to LCA of nanomaterials, several LCA studies related to glass fibre composites are also reported in literature. These studies have analysed environmental impacts of fibres and matrices used in the various applications of glass fibre composites. Results of these studies can help to understand the environmental impacts of nanoclay composites. Rosa et al. [126] reported that about 70% of environmental impacts of a composite component was driven by epoxy resin whereas glass fibre accounted for the remaining impacts. In another LCA study about glass fibres, it was reported that during production of glass fibres the energy used to melt the glass accounted for majority of carbon emissions associated with glass fibre production [127]. For this reason, replacing glass fibres with natural fibres has been investigated by various researchers as one of the most promising ways of reducing the environmental impacts of composites materials [126,128,129,130,131]

Overall, it is reported that using plant-based fibres in composites improves the sustainability of composites [132] and several types of bio fibres can be used for reinforcing composite materials. Oliver-Ortegar et al. [133] reported a study in which authors have used cellulose as a reinforcing nanofibre network in a polymer matrix and have reported that the resulting composite showed excellent mechanical properties. In another study [134], recycled cotton fibres from textile waste were used to substitute glass fibres in composites materials. The flexure strength analysis of this bio composite showed promising results and demonstrated the possibility of replacing glass fibres with cotton fibres, thus reducing the carbon emissions related to production of glass fibres. Chihaoui et al. [135] have reported a study in which they have used date palm waste to product nano fibres with diameter ranging from 10 to 30 nm. They reported that composites prepared using these fibres demonstrated enhanced tensile strength and Young’s modulus. In another study [136] it was shown that non-woven natural fibre can be used in developing bio-composites for industrial applications. These studies point that using bio fibres is a viable option to reduce environmental impacts of composite materials.

Due to impending threat of extreme climate change, there is a growing concern about environmental impacts of various novel materials such as nanocomposites. As the population is becoming aware of adverse effects of various technologies, demand for sustainable and green solutions is growing. Therefore, while introducing new materials it important to consider the environmental impacts of these materials. Sustainable materials should balance technical, economic and environmental constraints. By considering environmental impacts of novel materials in early stages of their design, negative environmental and human health effects can be avoided. Given that the demand for nanomaterials including nano composites is rapidly growing and that their environmental impacts are not fully understood, future studies should develop in depth methods to analyse various environmental impacts and human health issues caused by nanocomposites.

## 5. Conclusions and Recommendations

The addition of inorganic nanoclays to epoxy resin improves its physical, mechanical and barrier properties. The most common nano fillers used with epoxy resins are the organically modified montmorillonite clays with filling amounts ranging from 1% to 10%. Low clay contents are shown to result in better E-nc structures. It is proven by several researchers that E-nc with exfoliated morphology results in noticeable improvements in most of the properties of the nanocomposite as compared with the pristine epoxy. This E-nc morphology has been shown to maximize the interactions between the resin and the nanoclay platelets, resulting in appreciable improvements in the mechanical, thermal and liquid barrier properties of the epoxy. Improvements of more than 68% in tensile strength and 80% in resistance to water absorption were reported. The complete exfoliation of nano fillers in epoxy matrices remains quite a challenge because the degree of exfoliation depends on the type of epoxy, the type of organically modified clays, the clay loading and the mixing and curing methods. 

Because of its excellent mechanical strength under both static and cyclic loading conditions GFRE has found a large number of applications. One major obstacle to a wider use of GFRE in structures, such as piping systems, is the affinity of epoxy resin to moisture absorption, especially at high temperatures. The combination of the nanoclay together with glass fibres as reinforcements has been shown to lower the affinity to moisture uptake and improve the mechanical properties of GFRE. Addition of the right amount of clay improves the tensile, flexural and impact properties of GFRE. The maximum water uptake decreases with increasing clay loading and moisture diffusion at higher temperatures was higher than that at room temperature. Though exposure to moisture of GFRE-nc resulted in the degradation of its mechanical properties, they always remained comparably higher than the pristine epoxy.

There is almost consensus that properly dispersed and distributed nanoclays improve the physical, mechanical and thermal properties of epoxy. However, there are still contradictions on how much improvements and how to arrive at the complete exfoliation of nanofillers in polymer matrices. This requires an effort to establish detailed procedures for each type of organically modified clay and epoxy. The problem of nanoclay filtration during the production of GFRE-nc needs more investigation. Finally, like every novel material, epoxy nanoclay composites have environmental impacts that need to be fully understood. Further research is needed to develop environmental profiles for these materials. Special focus is required to incorporate end-of-life phases into LCA studies. Similarly, analysis of exposure pathways in various applications of these materials should be investigated to fully understand their effect on human health.

## Figures and Tables

**Figure 1 polymers-14-01620-f001:**
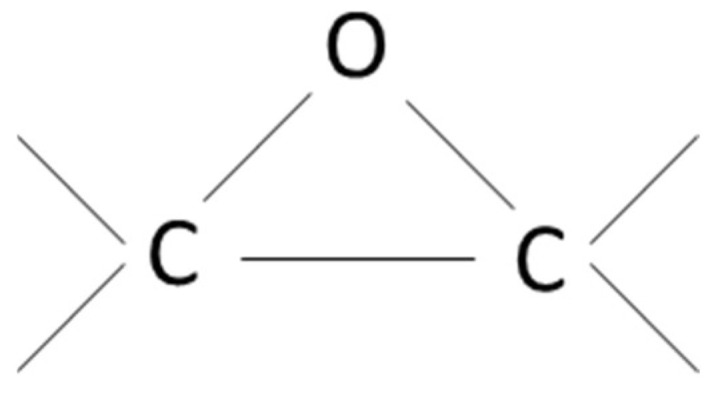
Epoxy group.

**Figure 2 polymers-14-01620-f002:**
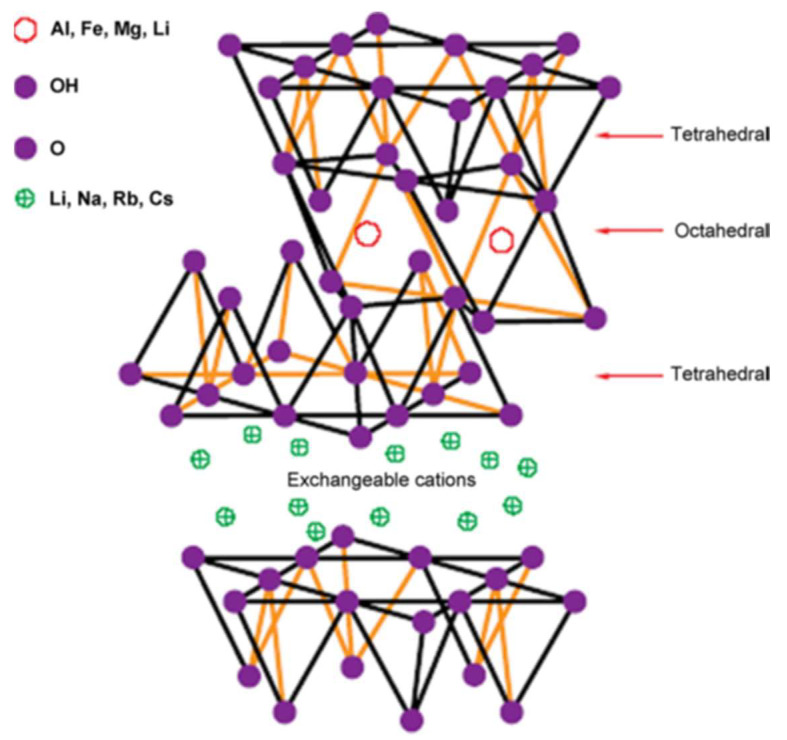
The structure of a 2-D layered silicate. Reproduced from [34] with permission from Elsevier. (Copyright 2022, Elsevier).

**Figure 3 polymers-14-01620-f003:**
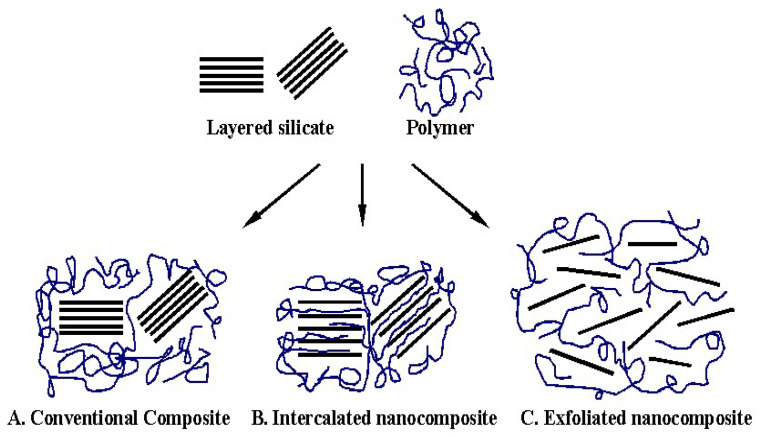
Structure of composites arising from the intercalation of layered silicates and polymers: Ref. [35]. Reproduced from [34] with permission from Elsevier. (Copyright 2022, Elsevier).

**Figure 4 polymers-14-01620-f004:**
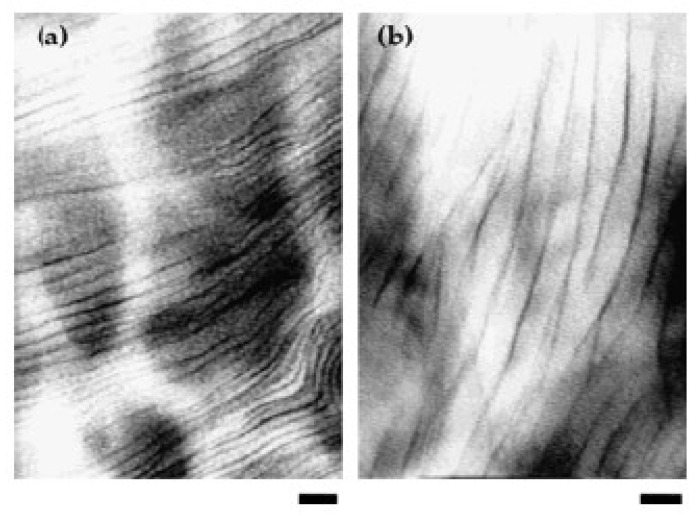
TEM images of (**a**) exfoliation and (**b**) intercalation structure of DGEBA-MDA and DGEBA-DDS systems respectively. Scale bar corresponds to 20 nm [63]. Reproduced from [63] with permission from ACS Publications.

**Figure 5 polymers-14-01620-f005:**
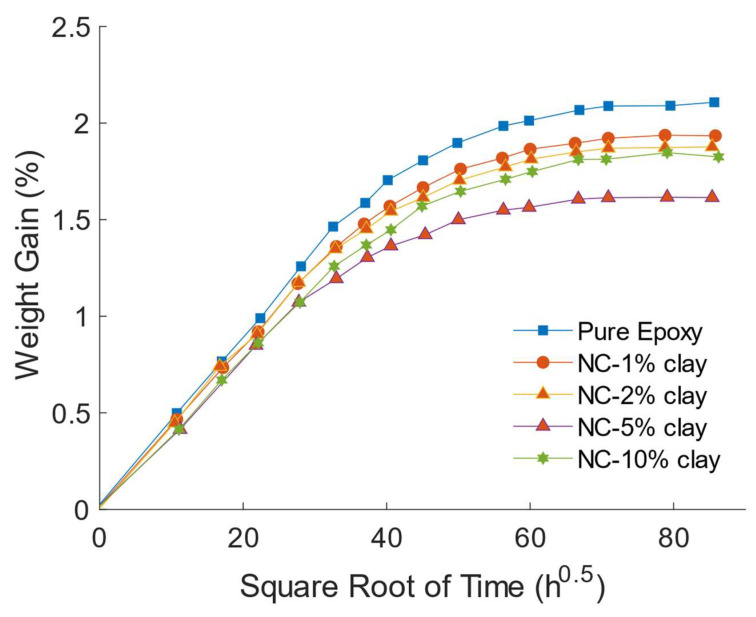
Variation of percent weight gain (water) with the square root of exposure time [29]. ((2013, Springer Nature).

**Figure 6 polymers-14-01620-f006:**
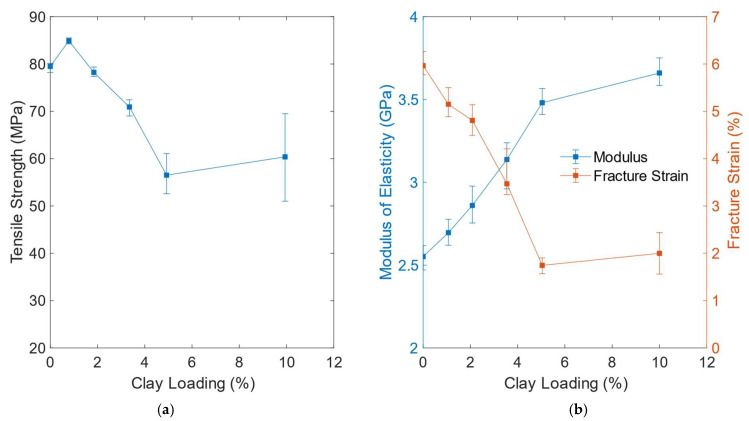
Effects of I.30E clay loading on the (**a**) the tensile strength and (**b**) the modulus of elasticity and fracture strain of DGEBA epoxy from [29]. (2013, Springer Nature).

**Figure 7 polymers-14-01620-f007:**
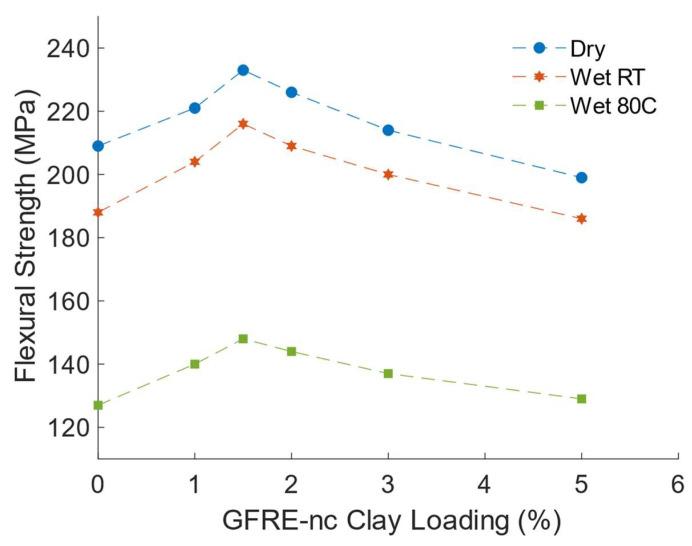
Effect of water uptake on flexural strength of GFRE and GFRE-nc at room temperature (RT) an 80 °C: (Zero (0) on the abscissa represents glass fibre reinforced epoxy with no clay addition).

**Table 1 polymers-14-01620-t001:** Comparison of tensile properties of pristine epoxies and nanocomposites with organoclay modifiers.

Sample	Tensile Strength, MPa (% Change)	Elasticity Modulus, GPa (% Change)	Fracture Strain, % (% Change)	Reference
Pristine DGEBA	80	2.54	6.02	[29,39]
Pristine EPON 828RS	62.5	3.03	7.8	[38]
2% I.30	79.3 (−0.9)	2.85 (+12.2)	4.8 (−20.3)	[39]
1% I.30E	60.6 (−3)	3.1 (+2.3)	5.9 (−24.3)	[38]
3% I.30E	69.7 (+11.5)	3.33 (+10.0)	6.4 (−17.9)	[38]
2%I.28E	55.4 (−24.6)	2.85 (+12.2)	2.49 (−58.8)	[39]
3% I.28E	57.5 (−8.0)	3.03 (0)	6.0 (−23.1)	[38]
2% C10A	49.8 (−37.5)	3.09 (+21.6)	1.83 (−69.6)	[39]
3% C10A	44.2 (−13.3)	3.08 (+1.7)	3.5 (−55.1)	[38]
2% C15A	77.9 (−2.6)	2.96 (+16.5)	4.56 (−24.3)	[39]
3% C15A	53.1 (−9.8)	2.96 (+2.3)	4.6 (−41.0)	[38]

**Table 2 polymers-14-01620-t002:** Effects of mixing technique and I.30E clay content on the tensile properties of epoxy.

Type of Organically Modified Nanoclay	Mixing Technique	Clay Loading (wt%)	% Change in Tensile Strength	% Change in Tensile Modulus	% Change in Fracture Strain	Reference
Octadecyl ammonium (I.30E)	High Shear Mixing	1	+7.0	+6.6	−13.6	[29]
		2	−0.5	+12.2	−21.9	
		3.5	−8.2	+18.1	−41.8	
		5	−28.4	+34.6	−70.1	
		10	−23.7	+40.9	−66.7	
		2	−0.9	+11.4	−18	[39]
		3	−16	+22	−45	
		5	−31	+31	−66	
		10	−29	+31	−63	
	Mechanical Stirring	2	−11.58	−11.19	−4.5	[75]
		5	−20.22	+3.9	−36.6	
		10	−23.75	+10.85	−40.4	
		1	−3.04	+2.29	−24.3	[38]
		3	+11.5	+10.0	−17.9	
		6	−10.72	+13.83	−52.6	
		10	−31.04	+20.06	−69.2	

## Data Availability

Not applicable.

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
