# Peer review of "Mechanical and Moisture Barrier Properties of Epoxy–Nanoclay and Hybrid Epoxy–Nanoclay Glass Fibre Composites: A Review"

_polymers, 2022, doi:10.3390/polym14081620_

Round 1
Reviewer 1 Report
Mechanical and physical properties of epoxy-nanoclay and hybrid epoxy-nanoclay glass fiber composites : A review
The paper is well written but lake of reference data and should stay focus on Epoxy and clay.
Epoxy is a versatile industrial product, same as clay. I would expect an introduction on epoxy families and clay families. Another approach would be to focus on a specific epoxy and clay family. For instance, reduction of the Tg reported in the conclusion should be more accurate. Which Epoxy, which clay?
Nanoclay is not introduced. What does mean nano-clay?? A clay is made of elemental particulates. Such particulates have a sheet-like structure. What is the lateral an thickness of these sheets ? Does clay mineral be a nano-mineral or the result of dispersing clay in an epoxy matrix be a nanoclay? (What are Cloisites or cloisites or nanomers ? (It seems to be trade names))
I reported several question or comment in yellow in the paper enclosed.
The first citation is too old and I would remove it.
The part concerning Environmental impacts of epoxy-nanoclay composites should be removed. It is too general. If the authors want to keep this chapter, epoxy and the environmental issue with this thermoset should be treated. Data on the carbon dioxide emission of epoxy, epoxy-nanoclay and epoxy-nanoclay glass fiber production should be given. I have noticed that the authors did not conclude on this chapter in the general conclusion meaning that this part should be indeed removed.
Conclusion should be 10 times shorter with quantitative data. For instance, improvements of more than 68% in tensile strength and 80% in resistance to water absorption were reported. I cannot find those numbers in the text. The quantitative value of water absorption of epoxy should be given since it is the reference value. Adding clays reduce this reference value by how much %? Adding nanoclays reduce the value by how much? (idem for Glass Fiber Reinforced)
It is unclear if the mechanical properties are improved because of the presence of nanoclay and GFR or because of the decrease of water absorption… Epoxy as a plastic does not absorb too much water but this parameter seems central in the improvement of the mechanical properties

Author Response
Please see attachement

Reviewer 2 Report
The paper is a review on the "Mechanical and physical properties of epoxy-nanoclay and hybrid epoxy-nanoclay glass fiber composites". It is well organized and describes correctly the state of the art in the subject of the review.
The major drawback of the paper is that besides 3 general figures taken from the literature there is no comparison done in figures or in tables between the results obtained in various researches. Therefore the text becomes very difficult to be followed and even boring. To give just an example: in 2.4 as presenting the mechanical properties of epoxy nanoclay composites the reader cannot find any clear indications about the values of mechanical properties of these composites. A statement like "addition 3% of nanoclay loading in epoxy resulted in the highest tensile, flexural and impact strengths as well as fracture toughness" or "percentage elongation and Young’s modulus increased gradually with clay loading" is meaningless as long as there are not given any values for these properties.
Therefore, I recommend the complete rewriting of the paper. Read some review papers to see which is the format. Show in figures and tables how different properties are comparing between them, which improvements were done, and comment on which are the future developments in the domain.
Reviewer 3 Report
The manuscript "Mechanical and physical properties of epoxy-nanoclay and hybrid epoxy-nanoclay glass fiber composites: A review" is a very extensive study. The authors refer to 132 items in it. After reading the manuscript several times, I believe that it is correctly written, but in practice it does not bring anything revealing. In fact, the authors only mention achievements in the field, but without the deeper aspect of basic research. In the title, the authors write about "physical properties", but this is just an empty phrase. The article does not introduce a significant degree of the physical phenomena that are the basis of the mechanical properties of epoxy-nanoclay and hybrid epoxy-nanoclay glass fiber composites (which, in my opinion, should be the aim of the study).
I have no technical objections to the manuscript (except for errors in references). However, I believe that the scientific level of the work is too low for publication in Polymers and it does not bring anything new to the research on polymers or composites.
Round 2
Reviewer 1 Report
Dear authors and collegues,
I hope that my requests for modifications were not too painful and that you genuinely believe that these modifications were necessary to improve the understanding of your works.
Best regards
Reviewer 2 Report
After the corrections and additions done in the revised form, paper can be published as it is.
Reviewer 3 Report
The additional figures, tables and descriptions significantly improved the quality of the manuscript. I believe the article has been greatly improved and may be published in Polymers.